# Survey of Localization for Internet of Things Nodes: Approaches, Challenges and Open Issues

**Sheetal Ghorpade** [1], **Marco Zennaro** [2,*] **and Bharat Chaudhari** [3]

1   RMD Sinhgad School of Engineering, Savitribai Phule Pune University, Pune 411058, India; sn_ghorpade@yahoo.com
2   Science, Technology and Innovation Unit, International Centre for Theoretical Physics, 34151 Trieste, Italy
3   School of Electronics and Communication Engineering, MIT World Peace University, Pune 411037, India; bsc@ieee.org
*   Correspondence: mzennaro@ictp.it

**Abstract:** With exponential growth in the deployment of Internet of Things (IoT) devices, many new innovative and real-life applications are being developed. IoT supports such applications with the help of resource-constrained fixed as well as mobile nodes. These nodes can be placed in anything from vehicles to the human body to smart homes to smart factories. Mobility of the nodes enhances the network coverage and connectivity. One of the crucial requirements in IoT systems is the accurate and fast localization of its nodes with high energy efficiency and low cost. The localization process has several challenges. These challenges keep changing depending on the location and movement of nodes such as outdoor, indoor, with or without obstacles and so on. The performance of localization techniques greatly depends on the scenarios and conditions from which the nodes are traversing. Precise localization of nodes is very much required in many unique applications. Although several localization techniques and algorithms are available, there are still many challenges for the precise and efficient localization of the nodes. This paper classifies and discusses various state-of-the-art techniques proposed for IoT node localization in detail. It includes the different approaches such as centralized, distributed, iterative, ranged based, range free, device-based, device-free and their subtypes. Furthermore, the different performance metrics that can be used for localization, comparison of the different techniques, some prominent applications in smart cities and future directions are also covered.

**Keywords:** Internet of Things; IoT; localization; range-based; range-free; device-based; device-free; metrics; smart city

## 1. Introduction

The Internet of Things (IoT) allows the connections of a large number of sensors, actuators, and smart devices for persisting connectivity. Localization is an essential process in the IoT environment for tracking and monitoring the targets with the help of sensor nodes. The sensor nodes collect the target information and transfer it to the central controller for further processing. These applications demand information about the position of the sensor node, which is also essential in routing and clustering. The location of a node is generally determined by using geometric measures like triangulation and trilateration. The distance between any two nodes is determined using radio signal strength, coordination, important features of the resonant waves, and others. Localization approaches in wireless sensor nodes are independent of earlier location specifications; they rely on the location information of some particular sensor nodes and the internetwork measures [1]. The sensor nodes providing prior location information are known as anchor or reference nodes whose location is determined by the global positioning system (GPS). Recently, researchers have started studying localization in IoT networks for numerous applications [2–16]. However, all the sensor nodes cannot be equipped with GPS due to the cost, energy efficiency and

GPS signal unavailability in certain environments. Consequently, different approaches based on using a minimum number of anchor nodes to locate the other sensor nodes by information exchange among sensor and anchor nodes are proposed in the literature [2].

Node localization algorithms are mainly categorized as range-based localization and range-free localization. The range-based method utilizes hop distances, hop counts and angles for a position estimate. In contrast, the range-free method is based on the connectivity or pattern mapping for location approximation. Hybrid localization approaches developed by combining various range-based methods are precise and provide improved coverage [3]. Range-based schemes are partitioned into four categories by considering deployment scenarios; stationary sensor and stationary anchor nodes, stationary sensor and moving anchor nodes moving sensor and stationary anchor nodes, moving sensor and moving anchor nodes [4–6]. Device-based and device-free technologies are used for the localization of IoT nodes. Device-based technologies have progressed exceptionally towards the optimal location approximation, whereas device-free technologies are well suited for various application scenarios. Although various localization techniques are available for solving position estimate problems in the IoT networks, there are practical limitations in combining these techniques and deploying the minimum number of anchor nodes in such setups. Therefore, it is essential to design cost-effective and appropriate localization schemes. The motivation behind this review is in the study of distinct localization techniques and their applications in IoT environments.

In this paper, we present an in-depth review of range-free and range-based localization techniques and related concepts. The rest of the paper is organized as follows. Section 2 focuses on the classification of localization approaches. Section 3 presents the measurement techniques used for localization, covering the range-based and range-free techniques and their subtypes. Section 4 discusses device-based and device-free localization methods. We also discuss the crucial problems and fundamental challenges, and technologies. In Section 5, important smart applications such as smart city, health and industry, where localization plays a crucial role, are presented. Section 6 illustrates and compares the various metrics used in the performance evaluation of localization techniques. The paper is concluded in Section 7.

## 2. Localization Approaches

Based on the distance among the sensor nodes, localization techniques in IoT-enabled wireless sensor networks are classified into three categories: centralized, distributed and iterative. Classification of localization techniques is presented in Figure 1.

### 2.1. Distributed Localization Approaches

In distributed localization approaches, every node in the network shares its information with the adjacent nodes and estimates the distance for its position without involving the central unit [10]. Generally, distributed approach deduces the nodes' locations from the positions of the anchor nodes. The anchor node may have GPS capabilities to find its own locations. Hence, the sensor nodes should be directly located in the comprehensive coordinate system of the anchor nodes. Distributed approaches are more efficient and well suited for complex networks due to the involvement of every node in the process of the location estimate. Distance vector (DV) hop [11], DV distance [12] and other algorithms [13] are examples of distributed localization that utilize connectivity measures to evaluate the locations of non-anchor nodes.

The DV hop algorithm begins with the dispersal of all the anchor nodes through their positions with respect to the other nodes in the network. While broadcasting a message from one hop to another, every node preserves the information and determines the minimum number of hops needed to locate the anchor node. Whereas, DV distance algorithm focuses on broadcasting the estimated distance among the adjacent nodes rather than a number of hops in the process of the location estimate. The distributed approach correspondingly affords a degree of flexibility to the network and is also immensely

suitable for movable nodes since their position changes according to the requirements. The centralized approach is suitable for smaller size networks, whereas distributed approach is chosen for larger networks. The centralized and distributed approach in the IoT-enabled wireless sensor network is depicted in Figure 2.

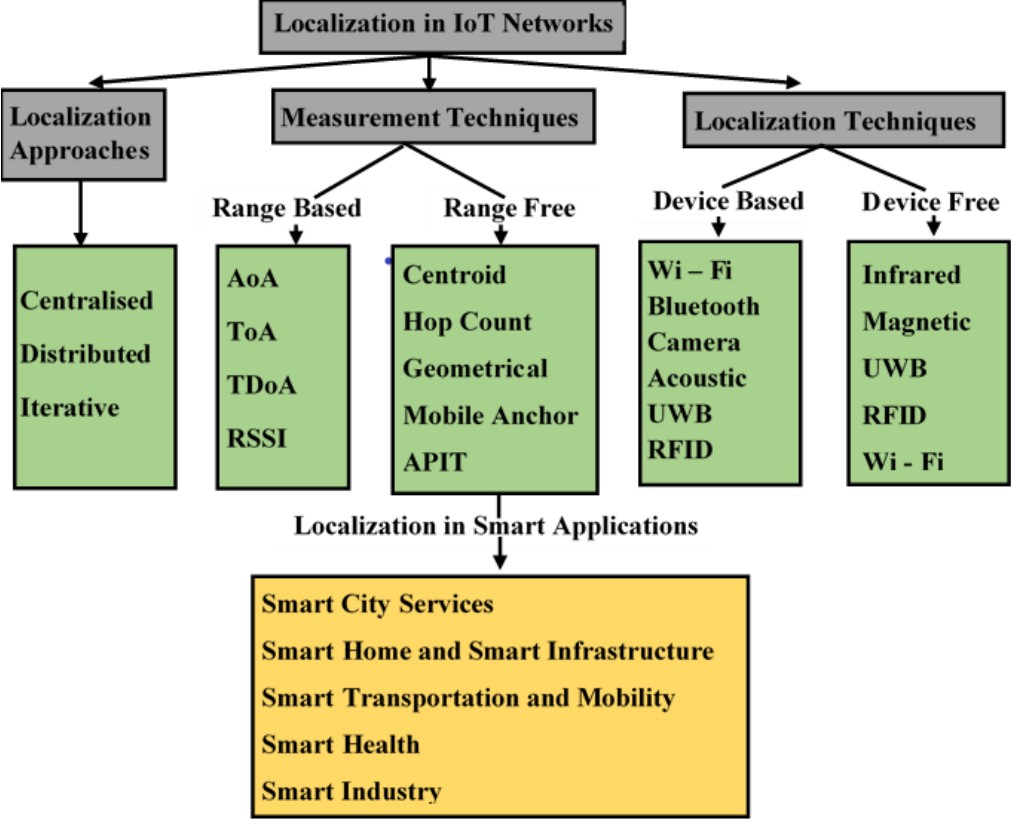

**Figure 1.** Classification of localization techniques.

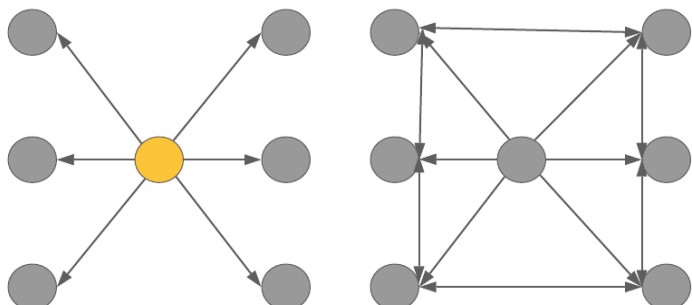

**Figure 2.** Centralized and distributed localization.

### 2.2. Centralized Localization Approaches

The centralized location approaches are based on powerful nodes for coordination among the nearby sensor nodes. The sensor nodes collect information such as received signal strength, details of adjacent nodes and then communicates it to the centralized node. The central coordinating node analyses the information and estimates positions of different sensor nodes using multi-dimensional and stochastic algorithms [7,8] and then conveys it to the individual nodes. Centralized localization techniques overcome insufficient resources at the sensor nodes at the expense of higher communication costs. With the increase in the number of IoT nodes in the network, centralized approaches have become highly expensive. The nodes closer to the base station get exhausted due to more involvement

in the communication process. The complexity of the centralized algorithms is high, and hence the individual nodes are not allowed to do location estimates. Multi-dimensional scaling (MDS) [7], stochastic optimization algorithms [8] and linear programming [9] are the main techniques used in the designing of these algorithms.

Centralized routing is appropriate for the networks in which the processing control trusts mainly on a single device. In these scenarios, the central device is accountable for processing, coordinating and managing the identified data activities. It also forwards the data to a sink node. The key advantages of the centralized routing technique are that it allows roaming inside the network and provides energy management and context information availability permits an improved application design in terms of nodes placement, application awareness, etc. Whereas, in distributed routing scheme, the information is managed by every node, and decisions are taken locally. The distributed routing included autonomous devices, and every node shares information to its adjacent node to have robust routing features. Therefore, it is good for larger networks and distributed applications such as multi-agent and self-organized systems.

### 2.3. Iterative Localization Approaches

This approach creates network topology with smaller network elements by dividing the network [14]. A network element can be either an individual node or a cluster of nodes, and every element possess its precise coordinate framework. Blending is indispensable in this approach and can be applied recursively to share the coordinate framework and generate more vital elements. Once the integrated coordinate framework is shared, all the nodes in the network get localized. The iterative localization approach helps in avoiding the local minima through the dimension reducing procedure. The distributed localization approaches can be enhanced by using the iterative mechanism for improving preliminary location precision. Iterative Cooperation DV (ICDV) hop is a distributed localization approach that follows the iterative mechanism. It chooses an optimum number of anchor nodes for higher localization precision and utilizes the hop threshold to restrict distance among the nodes.

Performance of centralized, distributed and iterative approaches can be analyzed by considering parameters such as position estimate accuracy, node density, design complexity computational cost and energy proficiency. Table 1 shows the performance comparison of these approaches.

**Table 1.** Performance Comparison of Localization Approaches.

| Localization Approach | Accuracy | Node Density | Design Complexity | Computational Cost | Energy Efficiency |
|---|---|---|---|---|---|
| Centralized | High | Low | Low | High | Low |
| Distributed | Low | High | High | Low | High |
| Iterative | Moderate | High | Moderate | Low | High |

Compared with centralized localization, the distributed localization approach is well suited for high-density networks and is computationally effective. Nevertheless, the IoT networks designed for health, agricultural and environmental and traffic regulation are based on centralized data collection architectures. In these frameworks, the information from individual sensor nodes has to be gathered and administered centrally, and hence the individual sensor nodes possess limited processing ability for energy saving. From the analysis of location estimate accuracy, it has been observed that centralized approaches generate precise estimation outcomes than the distributed, as they have the global perception of the network. On the other hand, higher computational complexity and irregularity due to losing information over multi-hop are pitfalls of centralized approaches.

On the contrary, designing distributed algorithms is more complex than centralized due to the local and global behavior difficulties. In specific scenarios, distributed algorithms provide the locally optimum solution but fail to optimize globally. Error in distance

estimation between sensor nodes propagated to other nodes further deteriorates the distributed algorithm's estimation accuracy. For calculating the energy consumption, the energy required for processing, conveying and receiving in the particular hardware and the transmission range must be considered. Due to the large number of communications in centralized approaches, energy consumption is higher than the distributed approaches.

On the other hand, distributed approaches are comparatively energy-efficient [15]. It can be observed that any distributed framework can pertain to a centralized one. Moreover, distributed forms of centralized algorithms can also be designed for specific applications. Such algorithms can have optimal trade-offs among the centralized and distributed localization approaches.

## 3. Measurement Techniques

The localization algorithms for IoT networks depend on a variety of measurement techniques as discussed in in this section. The important factors that affect the precision of localization estimate include the network topology, node density, number of anchor nodes, synchronizing timing, bandwidth and the geometrical dimensions. Furthermore, sensor nodes in the network can be static or movable, and their locations are determined either by using absolute coordinates or corresponding to anchor nodes. Localization algorithms may use 2-dimensional (2D) or 3-dimensional (3D) coordinates. The measurement techniques for localization of IoT nodes are broadly divided into two categories, viz. range free and range based. These two types of measurement techniques are explained in this section.

### 3.1. Range-Based Technique

In range-based algorithms, node locations are estimated by considering point-to-point distance or angle between the node with some reference. Some important approaches such as angle of arrival, time of arrival, time difference of arrival and received signal strength indicator are discussed in the section.

#### 3.1.1. Angle of Arrival

The angle of arrival (AoA) measurement technique is called the path of arrival or orientation measurement. The AoA is determined either by using the amplitude response of receiver antennas or the phase response of receiver antennas [16]. The angle is calculated when the maximum signal emerges from the anchor node to the unknown sensor node. An unknown sensor node's location is considered a line that makes a certain angle with the anchor node, as shown in Figure 3. Therefore, it requires at least two anchor nodes for determining the position of the unknown node. In this technique, a measurement error occurs due to multiple paths or shadows, leading to higher error in localization [16].

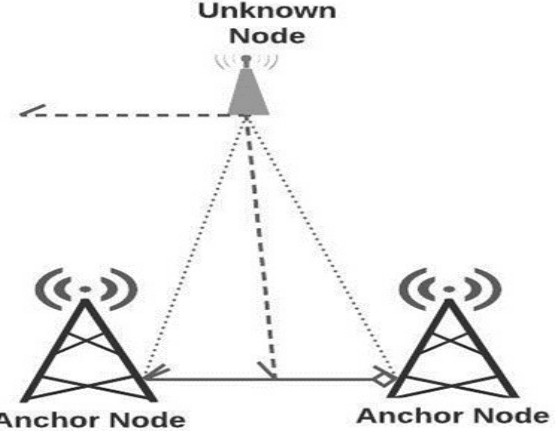

**Figure 3.** Angle of Arrival.

To attain desirable precision, larger antenna arrays must be used. Consequently, it demands added hardware with increased power consumption. Therefore, this technique is of limited interest, and improvements are expected for achieving optimal and feasible solutions in real-time application scenarios [17,18].

### 3.1.2. Time of Arrival

The time of arrival (ToA) measurement technique uses the propagation time, i.e., the time required for the signal to travel from the unknown node to the anchor nodes. The unknown node is to be in the range of anchor nodes. This technique requires at least three anchor nodes for determining the position of the unknown node. The assessed location of the unknown node falls inside the intersection area of the three circles, as represented in Figure 4. The realistic assessed location can be determined using the least square or weighted least square method [19].

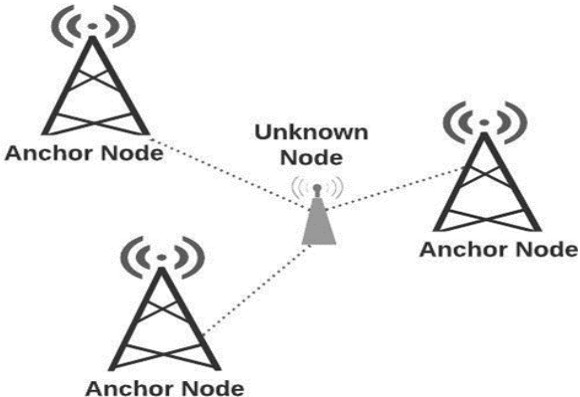

**Figure 4.** Time of arrival.

The technique requires accurate synchronization among the anchor nodes and unknown nodes, impacting the geolocation structure complexity. A system can use ToA measurements to overcome this drawback, built on a sound signal produced by the anchor node and at least four unknown nodes [20]. It improves the localization precision considerably. An enhanced ultrasound-based time of arrival technique [21] permits calculating the position in three dimensions and provides the data orientation. Cricket, a ToA-based localization approach [22], uses ultrasound transmitters with known locations. The anchor node evaluates the location of the unknown node with high accuracy.

### 3.1.3. Time Difference of Arrival

The time difference of arrival (TDoA) technique is based on the variations in the signal arrival time. The transmitter must be positioned on a hyperboloid in a TDoA measurement setup with the continual difference in the range among any two measuring components. These types of measurement are withdrawn from distinct pairs of reference points with knowledge about their positions. As well, instead of absolute time dimensions at every receiving node, corresponding time dimensions are utilized. In this technique, synchronization of time source is not required, but receivers must be synchronized. TDoA measurement techniques is also known as multilateral. The estimated position lies in the intersection of multiple hyperbolic curves, as represented in Figure 5.

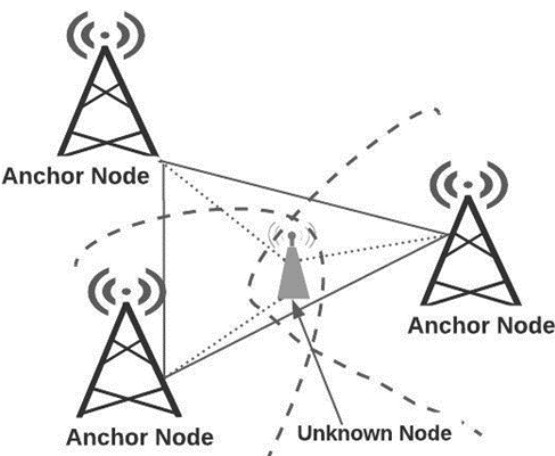

**Figure 5.** Time difference of arrival.

Precision is dependent on environmental parameters such as temperature and humidity along with synchronization error. If the source and receiving nodes are well gauged before communicating, it offers decent precision. However, the TDoA technique is not well suited for low-power sensor network devices [23,24].

3.1.4. Received Signal Strength Indicator

The received signal strength measurement technique uses the path loss log-normal surveillance framework to infer trilateration or uses the received signal strength fingerprints [25]. The primary method estimates the distance between the anchor node and the unknown node. Then it uses trilateration for estimating the position of the unknown node with the help of three anchor nodes. The latter method gathers the RSS patterns of the scenario, and the online dimensions are matched with the nearest probable position corresponding to the dimension database to determine the position of the unknown node, as depicted in Figure 6. RSS is a cheaper measurement technique that needs smaller hardware, but its location precision is sensitive to multipath broadcasting of radio signals [26].

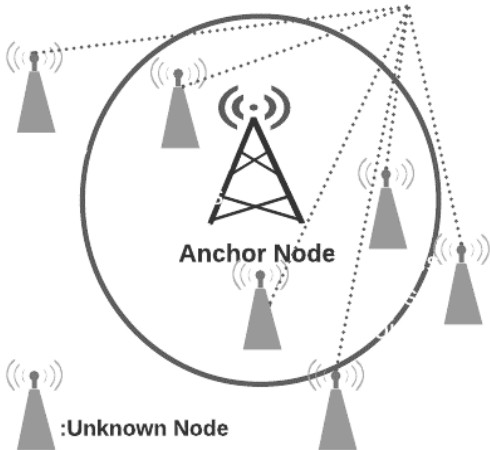

**Figure 6.** Received signal strength indicator.

Generally, independent localization approaches face an issue of precision. Subsequently, blending AoA, ToA, TDoA and RSS has been proposed to enhance position estimate precision [27]. The hybrid approach developed in [28] combines TDoA with RSS and improves position estimate precision. RSS measurement based on a fuzzy logic algorithm enhances the localization precision. A novel multi-objective optimization (MOO) agent-based on particle swarm grey wolf optimization (PSGWO) and inverse fuzzy rank-

ing is proposed in [29]. Initially, an enhanced PSGWO model is developed, and then it is utilized to develop population and multi-criteria based soft computing algorithms. This bio-inspired optimization technique determines the optimal path with minimum energy consumption and transmission cost in IoT networks.

*3.2. Range Free Technique*

Range free localization techniques do not require distance measurements, they utilize network data for a location estimate. They entirely depend upon the content of the received packet and have a lower cost than the range-based techniques [30]. In this approach, localization is based on geometrical interpretation, constraints minimization and region development, and it is a straightforward, economical and energy-efficient technique. We discussed important range-free techniques, viz. centroid, hop count, analytical geometry, mobile anchor node and an approximate point-in-triangulation test below.

### 3.2.1. Centroid-Based

The center of the geometric structure is calculated by finding the average of all the points those who generated the structure defined as

$$\left(u_{pred}, v_{pred}\right) = \left(\frac{u_1 + u_2 + \ldots + u_P}{P}, \frac{v_1 + v_2 + \ldots + v_P}{P}\right) \tag{1}$$

where $P$ represents a total number of sensor nodes and $\left(u_{pred}, v_{pred}\right)$ are an unknown node's coordinate. Most of the approaches in this technique are dependent on the center and the anchor node coordinates for estimating an unknown node's position [31].

Node density in the network does not affect the performance. The computational complexity of centroid-based techniques is $O(n)$, which is very low, and it achieves comparatively better precision in networks with uniform anchor node distribution. However, for the random distribution of anchor nodes, accuracy is very low. In weighted centroid localization (WCL) [32], initially, the anchor node communicates its position to all other nodes, then they determine their position with respect to the centroid. In WCL, weights are used to improve the calculations in which the weight is a function dependent on the distance and the receiver's features. Thus, it is reliant on communication range, and it does not require any extra hardware.

### 3.2.2. Hop Count-Based

Hop count-based methods are most popular in range-free localization techniques. The number of anchor nodes required in this technique are comparatively less. Distance vector hop (DV-Hop) plays a vital role in several localization algorithms by providing information about the initial distance among the sensor node and anchor node. Initially, anchor nodes broadcast hop count to the adjacent nodes, then every node collects the message updates the count. Nodes possessing higher counts are generally ignored. At the end of this stage, every node in the network must have a minimal number of hops through the anchor node then the anchor nodes estimate the mean size of every hop. The Hop count-based localization scenario is presented in Figure 7. Lastly, every sensor node in the network multiplies this mean size with the hop count to determine its distance from the anchor node.

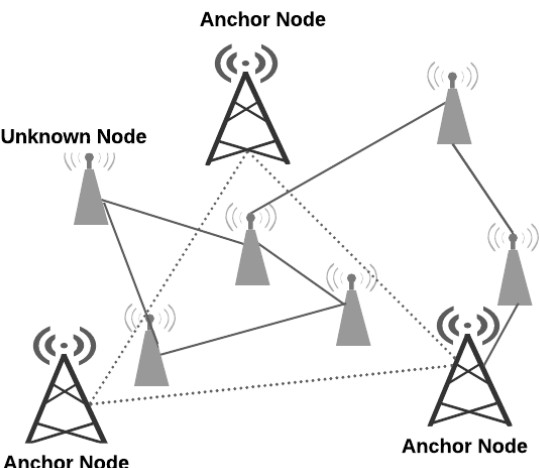

**Figure 7.** Hop count scenario.

Hop size between any two anchor nodes $(u_l, v_l)$ $and$ $(u_m, v_m)$ is determined by using

$$Hopsize_l = \frac{\sum_{m \neq 1} \sqrt{(u_l - u_m)^2 + (v_l - v_m)^2}}{\sum_{m \neq 1} N_{lm}} \tag{2}$$

where $N_{lm}$ represents the number of hops between the anchor node $l$ and anchor node $m$. DV-hop is a useful and straightforward localization technique. However, the drawback is that it needs homogeneously positioned IoT networks and the uniform amplitude reduction of signal strength in every direction. To overcome this drawback, improvements have been proposed in the literature [33–35]. Minimal mean square error measure for modifying mean hop distance [33] works well for isotropic topologies and improves location precision. However, for non-isotropic and anisotropic environments, this modification generates more errors while estimating the distance. Four closest anchor nodes assume that the shortest paths to the closest anchor node are less influenced by irregularities in the topology [36]. It produces better results for certain scenarios, but there is the probability of incorrectly discarding few virtuous anchor nodes to improve the location precision.

### 3.2.3. Analytical Geometry

The analytical Geometry-based technique uses statistical features of the network to evaluate the network's mean hop distance. Evaluated mean hop distance is locally assessable at every sensor node, and it has to be propagated to other sensor nodes. The localization algorithm for anisotropic environments proposed in [37] developed two approaches to compute the assessed distance among anchor nodes and sensor nodes; the first one is for the normally deviated, and the another is for the largely deviated anchor nodes from the sensor nodes. The normally deviated anchor node uses the data from the closest anchor nodes within three to four hops from the sensor node. Therefore, it requires a higher density of anchor nodes. The angle of the largely deviated path among the anchor and sensor nodes is determined and used for localization. Mean hop distance and hop count are not enough to determine the exact location of the sensor node [38]. It is also dependent on the number of nodes required for forwarding the information among any two nodes. The authors combined this information with the conventional data and achieved improvement in the localization.

### 3.2.4. Mobile Anchor Node

In the mobile anchor node-based technique, a mobile anchor node with GPS ability keeps moving into a sensing region and intermittently broadcasts its present location. The remaining sensor nodes gather the position coordinates of the mobile anchor node.

Afterward, the sensor nodes select three non-collinear coordinates of the mobile anchor node and apply different mechanisms to estimate position. A geometric conjecture-based localization approach with a mobile anchor node is proposed in [39]. The adjacent sensor nodes follow up of coordinates incoming and leaving anchor nodes to build an arc within its communication range. This process continues until the sensor node identifies at least three coordinate mobile anchor nodes, and then two chords are built among the identified points. Finally, the perpendicular bisector of these two chords generates the location approximations of the sensor node. This approach is further enhanced by which the intercept of any two identified coordinates regulates the limit region of the sensor node [40]. A similar process is continued for the other two pairs to reduce the area of the limit region of the sensor node. Then the mean value of all the intercept points gives the location estimate. Finally, a restricted area-based localization approach that uses a moving anchor node is proposed in [41]. Trajectories of moving nodes generate a particular type of restricted region for the sensor node. To recognize the probable position of the sensor node inside distinct limit regions, multiple intercepts are generated till the ultimate position is attained. However, an arbitrary waypoint motion model for the mobile anchor node generates higher localization error, and its computational complexity is also high. Geometric curve constraint approximation for the localization algorithm [42] uses the approximated constraints to create the chord on the virtual circle. The perpendicular bisector of the chords and approximated radius estimate the location of the sensor node. This algorithm also enhances the accuracy for borderline nodes. Though several algorithms have been developed using this technique, the major drawback is that they are not well suited for extensive periodic intervals and the unbalanced radio broadcast framework.

### 3.2.5. Approximate Point-in-Triangulation Test

The approximate point-in-triangulation test (APIT) is a region-based localization technique. In this approach, the network is divided into triangular areas among the anchor nodes. The unknown sensor node selects three anchor nodes who have sent the message to it, and then it examines whether it lies inside the triangle generated by the three anchor nodes or not, as shown in Figure 8. The selection process continues with distinct sets of three anchor nodes until all probable anchor triangles are generated or the expected precision is attained [43]. APIT algorithm works very well for the IoT networks with irregular sensing regions with random node locations and expects lower communication costs.

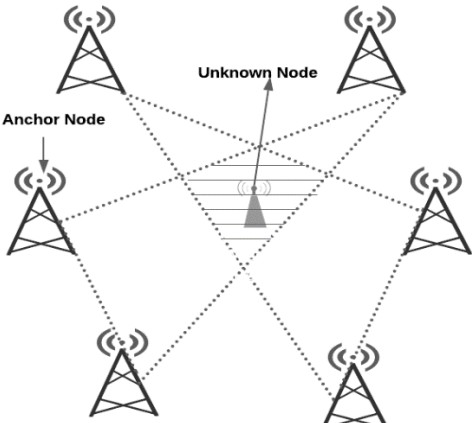

**Figure 8.** Approximate point-in-triangulation test.

A comparison of range-based and range-free localization techniques based on evaluation metrics is given in Table 2.

**Table 2.** Comparison of Measurement Techniques.

| Measurement Technique | Type of Algorithm | Accuracy | Cost | Energy Efficiency | Scalability |
|---|---|---|---|---|---|
| Range-based | AoA | Medium | High | High | Complex |
| | ToA | High | High | High | Complex |
| | TDoA | High | High | High | Complex |
| | RSSI | High | Low | High | Complex |
| Range-Free | Hop Count | Low | Low | High | Simple |
| | Centroid | Low | Low | Low | Simple |
| | APIT | Medium | Low | High | Simple |
| | Analytical Geometry | Low | Low | High | Simple |
| | Mobile Anchor Node | High | High | Low | Simple |

## 4. Localization Techniques

For IoT-enabled wireless sensor networks, localization is essential for indoor and outdoor surveillance services. A variety of localization algorithms has been developed in the last few decades. These algorithms use either device-based or device-free technologies, which are discussed in detail in this section.

### 4.1. Device-Based Localization

In this technique, specific devices like smartphones or tags possess the capability of providing desired localization information. A comprehensive study of smartphone and tag-based recent localization applications is presented here.

#### 4.1.1. Smartphone-Based Localization

Smartphone-based localization is a promising technology for solving localization problems in IoT applications. This approach is mainly classified into three types: Wi-Fi, Camera and Bluetooth [44].

#### 4.1.2. Wi-Fi-Based Localization

By detecting the Wi-Fi network, the device gets located. The device's location linked with the network is estimated by using identified positions of specific Wi-Fi networks. The framework of the Wi-Fi-based localization technique is shown in Figure 9. In this type of localization, accuracy depends upon the Wi-Fi access point capacity.

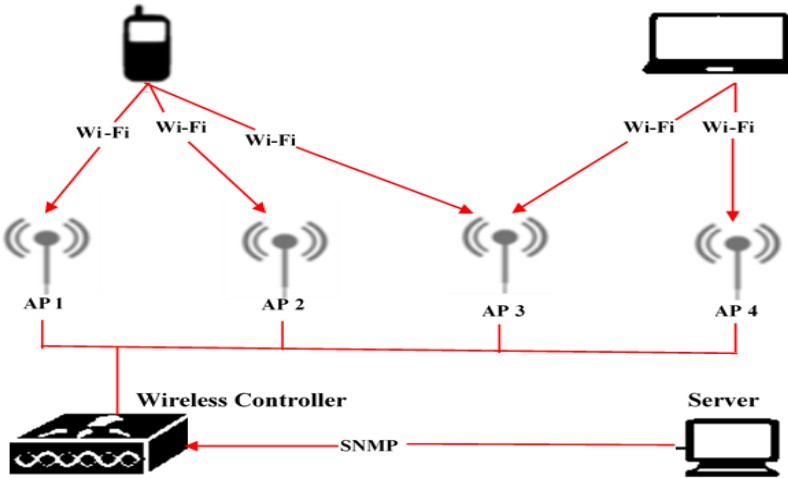

**Figure 9.** Wi-Fi based localization.

Wi-Fi fingerprint-based localization [45] uses the strongest access point for the maximum received signal strength (RSS) value. In the next phase, duplication of the access

points of the fingerprints and the distance generates accurate approximation irrespective of the building structure and data dispersal of the access point. The framework of the proposed approach is as shown in Figure 10.

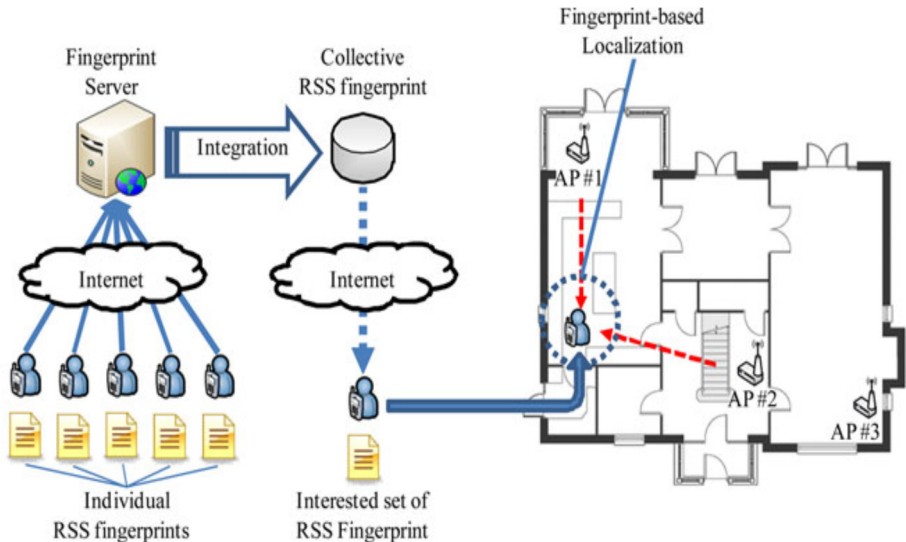

**Figure 10.** Fingerprint-based localization [45].

Triangulation ensures that broadcast constraints and AP localizations will be accessible in the initial stage itself [46]. Though complete data is offered, this real-time localization is not often suitable for environmental fluctuations. By merging RSSI received from the Wi-Fi and the inertial sensor dimensions of the smartphone to determine the location [47], it attains insistent location accuracy compared to the other methods.

### 4.1.3. Bluetooth-Based Localization

In this method, the location of a moving device is treated the same as that of an individual object. Location precision is dependent on the number and the size of the cells. Several localization schemes based on Bluetooth technology have generated enhanced accuracy. The location approximation algorithm proposed in [48] determines smartphone location with the help of RSSI measurement. The unified sensor node in the smartphone and the structure card request for the source data. Trilateration, established in [49], is used as a localization approach to the fingerprint to minimize earlier errors. Enhanced Bluetooth technology platform [50] for the remote-control application in an irrigation system which provides location information to the users.

### 4.1.4. Camera-Based Localization

In camera-based localization inside the building [51], the smartphone camera can be used as a sensor to computer vision for the position estimate. This technique combines distance approximation with image identification to attain the coordinates of the object to be positioned. To minimize the realization cost and for improving flexibility, unified sensor nodes are preferred. An optical camera and an alignment sensor node are proposed in [52], where adjacent nodes are identified with the help of fingerprints of the Wi-Fi signal. An improved localization method [53] features capture image and coordinates information to structure out fingerprints. This method works very well for indoor and outdoor environments and in regions with shadows.

### 4.1.5. Acoustic-Based Localization

Acoustic localization is a recently developed technique for location estimate which attains high accuracy with the help of a microphone and speaker into the smartphone. The indoor localization scheme [54] benefits from the audio input and output of the

smartphone and its refinement capabilities to execute audio modifications in the acoustic band dependent on non-interfering acoustic signals. Every smartphone can get its location to utilize all the audio signals by naturally synchronizing with the acoustic beacons. The time of arrival (ToA) measurement technique is used to detect distinct anchor nodes and developed an audio communication among the anchor nodes, loudspeakers, and microphones on a smartphone [55], and it improved the precision.

### 4.1.6. Tag-Based

Tag-based localization techniques require particular hardware set up for location compatibility. Ultra-Wide Band (UWB) and Radio Frequency Identification tag-based localization methods are discussed.

- Ultra-Wide Band Tag-Based Localization:

The larger bandwidth of Ultra-Wide Band (UWB) permits real-time requirements and improved secrecy. The smaller fundamental frequency consents to an upgraded wave channel via diverse materials. UWB tag-based localization algorithms have provided motivating outcomes. UWB tag integrated through the pulse transmitter intermittently transmits immensely smaller pulses at a precise rate. Accordingly, accepting the conveyed pulses is the responsibility of the base station [56]. Every moving anchor node in the network possesses a tag that communicates impulses to perceive automatically nearby the base station.

Additionally, this moving anchor node organizes and forms a self-organized wireless network. Besides, the high determination of UWB signals at various base stations can be reassembled jointly, trusting the density renewal procedure [57]. The synchronization of the clocks at the base stations is required so that the period variances of the impulses will imitate the geometrical alterations at other base stations. This process of localization is presented in Figure 11.

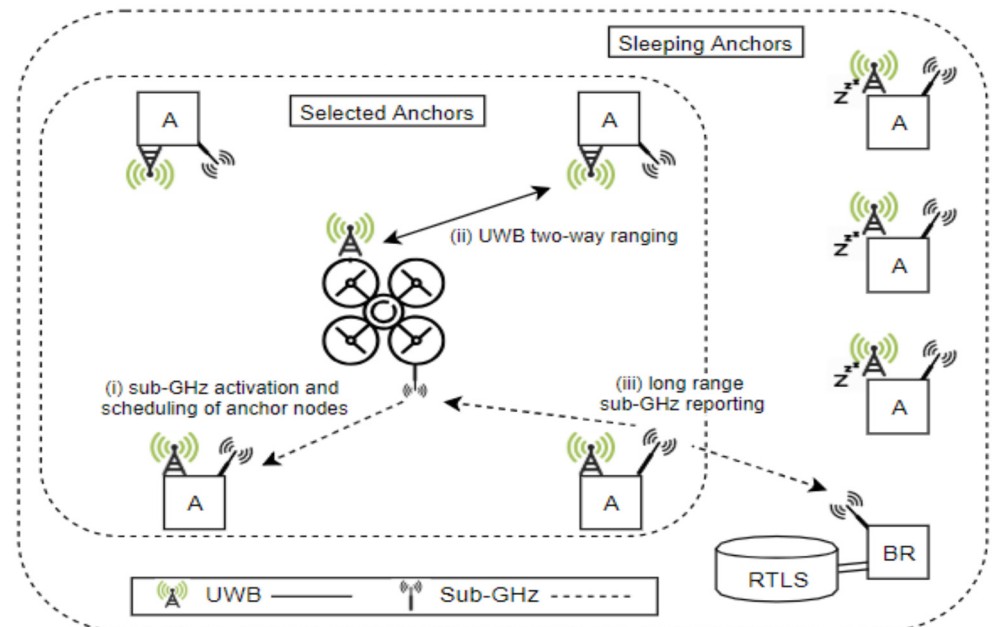

**Figure 11.** UWB tag-based localization.

- RFID Tags Localization

RFID tags integrated into an object can remotely identify, track, understand the features of an object. RFID technique permits construing tags deprived of direct view, and also it passes through the fine material color coatings, snow, etc. The RFID tag compromises

a chip linked to an antenna, enfold in support and interpreted by a reader, which seizures and communicates the information. Figure 12 shows the localization based on RFID tags.

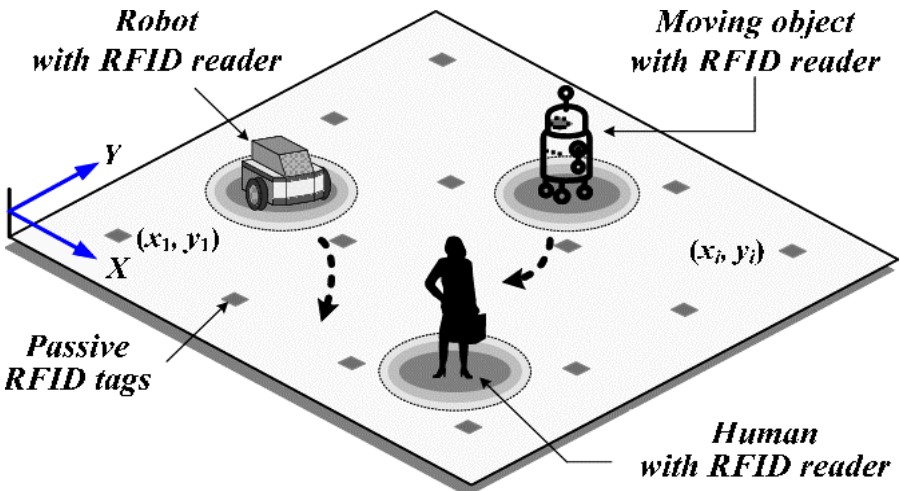

**Figure 12.** RFID tag-based localization.

RFID tags are mainly classified into three types: read-only/immutable, read-rewrite, and write once-multiple read tags. In the third type of tag, the chip has a blank memory region to write the specific number for the particular operator. Nevertheless, this number cannot be altered once it is embedded. Besides, RFID tags can be either active or passive. Active tags are associated with onboard power sources; they have an improved range, higher cost and constrained lifetime.

On the other hand, passive tags are cheaper, have an unlimited lifetime and require a substantial amount of energy disseminated at a smaller distance by the transmitter. In the localization approach for the objects in an office [58], every object carrying an RFID interpretation component can read passive tags mounted beside the object path. Coordinates of RFID readers are assessed by using received signal strength dispersed by the tags. RFID tags for position estimate in libraries and warehouses is proposed in [59] which equips all the moving objects with RFID chips. Tags are capable of locating objects crossing the range. This technique is economical, energy-efficient, and reference drive deployment is not required since the RFID reader must be moving continuously to scan the tags and location estimates. Tag resemblance and assembling in line with the distances measured by RSSI is used for localization [60].

*4.2. Device-Free Localization*

In this approach, the target localization can be achieved without integrating specific devices with the object to be tracked. Unfortunately, most device-free localization techniques depend on radiofrequency and object movements, disturbing the original radiofrequency patterns. In this section, we have discussed varieties of recently developed device-free localization approaches.

4.2.1. Infrared Localization

Infrared technology is broadly implemented for the recognition and localization of stationary devices. It is based on the radiation alterations in the series of infrared lights triggered by humans. There are apparent variations in an individual's body temperature and surrounding environment temperature utilized in the module localization scheme. An infrared-based crowd localization into the distributed wireless sensor networks [61], virtual and potent methods are used in parallel with correspondence with the angle bisector process of personal individual recognition and a scheme to group the dimension points. Positions are estimated by a filtering technique of the prime incoherent dimension

point. However, the involvement of a limited number of sensor nodes deteriorates the performance of this approach. To overcome this pitfall, we increased the number of personal individual recognition sensor nodes and fixed them to the ceiling of an area of localization [62]. Formerly, localized fixtures and the real-time information of moving objects has been used for crowd localization. Tracking accuracy by this enhancement has been improved up to 90%. A study shows that among three distinct methods; personal individual recognition (PIR) sensor dimensions, Ultra-Wide Broadband and RFID, the PIR and UWB perform better than RFID for indoor localization [63]. PIR sensor nodes and filters [64] are used to improve the accuracy of the indoor localization scheme, which is based on signal strength measures. However, this technique is not feasible in real-time scenarios, and also deployment complexity is high.

### 4.2.2. Magnetic Sensor-Based Localization

In magnetic sensor-based localization, the building plot is drawn without capturing GPS signal and by ignoring magnetic fields since advanced construction materials possess magnetic inscription features. An underground pipeline monitoring scheme developed in [65] perceives and traces leakages in concurrent time from distinct sensor nodes lying inside and around the pipeline. The manifestation that the peculiarities use the Earth's magnetic field for accurate wayfinding inspired the authors of [66]. It developed a localization technique for large buildings having long corridors connecting distinct regions. Local signals originating through the earth's magnetic field are used for location estimates. Chen et al. [67] used resident characteristics of the variable magnetic field to extant a physical amplitude in three-dimensional fruition environment realization. The magnetic field is a promising technology for geo-localization and optimal accuracy.

### 4.2.3. Ultra-Wide Band Radar Localization

UWB radar is a device-free localization technique comprised of transmitting and receiving nodes, and it is used for detecting and tracking a mobile object in the surveillance region. UWB localization scheme combines the frameworks of operator-based ultra-wide bands, whereas conventional schemes are built on energy detection [68]. Additionally, the proposed scheme permits determining object location in the microenvironment irrespective of the localization error. Localization of an interior mobile object using the transmitter and multiple receivers of the UWB radar is presented [69]. Particle filtering is also used for localization affected by canopy zones by focusing on the projected locations of the particles.

### 4.2.4. Radio Frequency Identification (RFID)

RFID is a technology that trusts radio waves for automatically identifying objects, and it is broadly used in shopping malls, hospitals, multiplexes, etc. An algorithm for object piloting by installing a cluster of RFID tags beyond the walls as an antenna array and monitored object's echoes with the help of concealed Markov module is developed [70]. It shows an average localization error of around 18 cm. RFID-based scheme for monitoring the people inside the building [71] uses inactive RFID antennas and radios. A neural network is integrated for improving location precision. Rauan et al. [72] developed RFID COTS, and inactive RFID tags are provided by radio signals and distribute the same data through a small dispersed signal. The biggest drawback of these RFID COTS is that they fail to read RSSI for moving objects and high-density locations.

### 4.2.5. Wi-Fi-Based Localization

Wi-Fi-based localization estimates the object's location by using particular features of the signal distribution. However, it is a comparatively expensive setup. Gong et al. [73] used the K-Nearest Neighborhood algorithm to develop wireless subarea localization (WiSal) positioning scheme to estimate the human location. It also uses signal modifications of distinct sub-regions through clustering. RSS on few antennas specifies the individual's accessibility to the receiver, which requires a massive difference in the signal magnitude.

Consequently, it helps to differentiate simply the human location into sub-regions. Channel State Information scheme [74] is resilient to sequential alteration and susceptible to environmental changes through frequency divergence. The advantage of frequency divergence in CSI is that it considers diverse multichannel replications. In the experimental phase of CSI, inactive fingerprints for the location estimate of a solo object are generated. Wi-Fi-based localization is most suitable for indoor localization. Device-based and device-free localization techniques are summarized in Table 3 with reference to the parameters like; accuracy, energy efficiency and cost.

**Table 3.** Summary of device-based and device-free localization techniques.

| Technique | Technology | Accuracy | Energy Efficiency | Cost |
|---|---|---|---|---|
| Device-based | Wi-Fi | Medium | Low | High |
| | Bluetooth | High | Medium | High |
| | Camera | High | High | High |
| | Acoustic | Medium | Medium | High |
| | WEB | High | Low | Medium |
| | RFID | Medium | Low | Medium |
| Device-Free | Infrared | Low | High | Low |
| | Magnetic Sensors | Low | Medium | Low |
| | WEB | High | Low | Low |
| | RFID | High | Low | Low |
| | Wi-Fi | Medium | Low | Low |

## 5. Localization in Smart Applications

One of the prime objectives of the IoT is to make our day-to-day activities more suitable by employing devices enabled with computational and communication proficiencies. The intelligible human interfaces, such as devices, transportation items, supply chain objects, etc., are becoming sovereign and multifaceted. It is reinforced by smart objects, possessing digital characteristics and compatibility to variations in the surrounding milieu. IoT solutions and principles have pertained to smart city setups for functioning and control [75]. A smart city is composed up of several components that are shown in Figure 13. Localization approaches for some of the smart components are discussed in the next section.

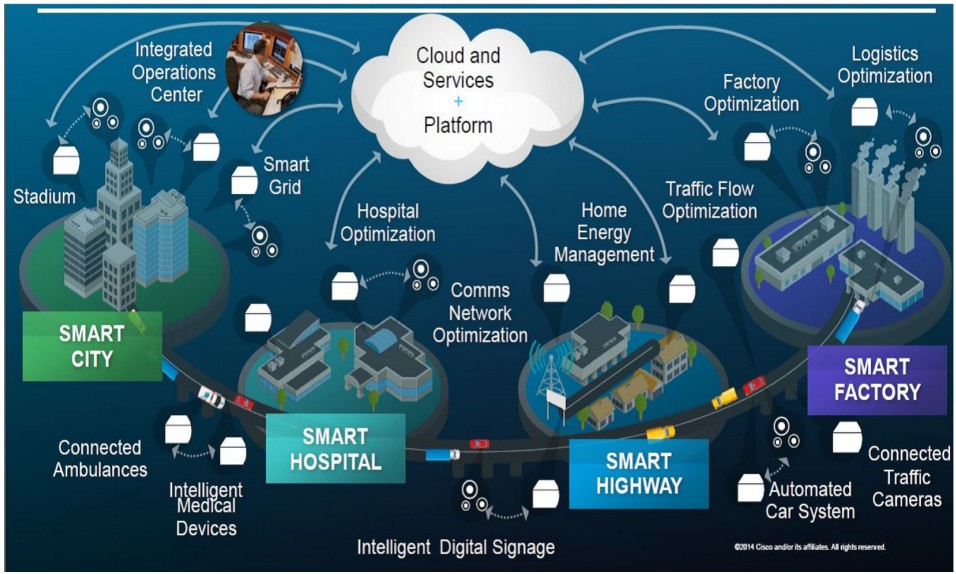

**Figure 13.** Major components of smart city.

### 5.1. Smart City Services

This section emphasizes the positioning-based services in smart cities. Position-based services integrate the location of a moving device into the other information to offer additional standards to the user. Consequently, the objective is to implement the IoT to enhance localization schemes and guarantee user-friendly processing in dynamic situations. Despite its dimensions, smart cities require continuous maintenance and cleaning by specialized machines and trained staff. In this context, the ultra-wideband (UWB) scheme proposed in [76] uses a semi-automatic floor scrubber adjunct component to clean indoor and outdoor tiles in smart cities. The accuracy of this localization scheme is around 20 cm. IoT nodes' detecting and networking capabilities help in optimal scheduling of energy dispersal and consumption in heterogeneous environments. The device that benefitted from these properties determines the location of the faulty parts, disconnects them and applies a swapping assignment to improve the major number of vigorous parts of the influenced energy flow. Furthermore, it uses a self-curing method capable of activating users' involvement and disseminated generation elements. A localization scheme using infrared technology [77] is developed, and it relies on the information gathered by the inactive infrared sensor nodes connected with the poles of light. An algorithm [78] localizes Wi-Fi admittance points and structures to regulate the urban noise sources. It does not require anchor nodes but gradually trusts crowdsourcing information to enhance the localization outcomes, aiming for improved precision.

### 5.2. Smart Home and Smart Infrastructure

A smart home is a residence that interlinks and controls smart devices to offer everyday activities of non-directional data to residents. It provides facilities and pertinence for upgraded health, ease and security. Few smartphone applications involve real-time user localization, which is especially imperative for elders and physically disabled people. Moreover, the localization permits smart devices to perform elementary tasks and executes complex commands through flexible interactions with user [79].

The indoor localization approach developed in [80] is accurate and is modified according to the environmental dynamics. It does not entail surveys and the primary training phase. The approach also captures interferences, broadcast turbulences and dimensional errors. In a smart home, all the gadgets in a house are supervised automatically and concurrently with respect to the resident's location. It includes communication among the localization scheme and the remote controller service area. Gadgets in the smart home execute inevitably with respect to in and out moves of the resident. It includes services like closing and opening shutters, security alarms, adjustments of humidity sensor nodes and radiators, automatic control of temperature and electricity, etc. In addition to this, it informs directly to the resident of any unusual observations.

Bluetooth is used in [81] for indoor localization in a smart building. Best accomplishment strategies applied in this scheme lead to a closer estimate of the object's location. The object distance improved by 22% and established 66% additional rewards compared to the supervised deep reinforcement learning model. The time-based localization scheme proposed in [82] uses Wi-Fi to evaluate the position of an object. It is structured to process smart devices, which smears the time intermission as the difference in the positions and the variable quantity of the Wi-Fi location to spot time points.

A positioning scheme based on Bluetooth technology to outline an association between the RSSI and the device location [83] utilizes the activities of a user for narrowing the exploration space iteratively to locate the desired object.

### 5.3. Smart Transportation and Mobility

Recent advances in linking vehicles to the Internet have given rise to ease and safety in transportation facilities. An idea of the Internet of Vehicles (IoV) connected with the Internet of Energy (IoE) characterizes imminent inclinations towards smart transportation and significance. Likewise, generating novel dynamic environments built on conviction,

safety, and accessibility for transport applications will guarantee customer concerning communications and services. Intelligent transport schemes have been extensively studied over the last few decades for providing advanced and motivated facilities for traffic organization and steering security concerns. Chen et al. [84] have developed a proximate vehicle map building procedure for localization, in which the accuracy is highly reliant on the location outcomes of every vehicle. Actually, maximum vehicles conveying approaches are omnidirectional, affecting the information sharing among the vehicles when their accurate location is not easily sensed. The parking lot and gate monitoring scheme proposed in [85] uses a wireless sensor network and active RFID. Gate monitoring is an economical and simple model in which RFID tags are assigned to subscribed users or provided dynamically at the entrance to the transitory users. Ghorpade et al. [86] developed a multi-objective grey wolf optimization-based model for optimal localization in smart parking. The optimization algorithm is used to minimize localization error. The objective functions have included the distance and topological constraints. ITS developed in [87] permits to trace disabled users. It affords flexibility by uniting stereo entity recognition with RFID and Bluetooth technology to improve pedestrians' operational abilities for interacting with transportation infrastructure. A cloud-based smart car parking scheme for smart cities uses the particular software progression method. It reduces disturbances and increases convenience and safety.

*5.4. Smart Health*

With the advancement of information technology, the idea of smart healthcare has progressively originated. Smart healthcare practices an emerging technology, IoT, big data, cloud computing and artificial intelligence to significantly revolutionize the conventional medical structure to make healthcare more competent, convenient and customized. Smart healthcare is a multi-layered change manifested from disease-centric to patient-centric care, from clinical automation to regional medical automation, from general administration to individualized administration and from concentrating on disease treatment to concentrating on preventive healthcare. These changes emphasize accomplishing personal requirements while improving the proficiency of medical care. It dramatically improves the medical and health facility experience and signifies the forthcoming progress track of modern medicine [88].

In this context, various IoT applications for localization in health care are developed by the researchers. Ha and Byun [89] developed a framework highlighting the situation cognizance trusting movable sensor nodes in a home care environment. The triaxial accelerometer detects the user's moves and incorporates radio allied with the Zig-bee network offers the location information using radio fingerprint technique.

IoT-enabled wearable sensor nodes can play a crucial role in monitoring the health and localization of senior citizens. These nodes can read vital health-related and other parameters, identify emergencies, and inform caretakers for immediate response. This approach includes a dive from scenario monitoring to constant perception and integrated care. It further reduces the associated risks caused by the disease while making it easier for medical institutions to monitor the prognosis of the disease [90].

A variety of approaches for wearable sensor nodes have been proposed for fall detection or indoor monitoring of the elderly in different scenarios. Chen et al. [91] have developed and applied the health care system for fall detection and localization of the elderly. A person's fall is detected by a mobile device and communicates to the help center via the ZigBee access point working in the 2.4 GHz band. It uses a triangulation method for localization and achieves 99% precision for fall identification. Fall detection for elderly persons and the ECG signal monitoring system is proposed in [92] for the outdoor environment. For improving the precision of the fall detection system, the ECG signal and GPS is employed. The ZigBee protocol is used to transmit the data to the centralized server and then to the healthcare center. An IoT-based range-based localization for smart city applications is proposed for accurate and low-cost localization. The extreme learning

machine (ELM), fuzzy system and modified swarm intelligence is used to develop hybrid optimized fuzzy threshold ELM (HOFTELM) algorithm for the localization of elderly persons in smart cities. The algorithm outperforms the existing algorithms in terms of average location error ratio (ALER) and is computationally efficient [93].

### 5.5. Smart Industry

Smart industry infrastructure can be deployed in diverse applications of manufacturing, production, supply chain, quality assurance, predictive maintenance and control, optimization of resources and others. It contains intelligent machines, robots, equipment and tools with multiple IoT sensors to monitor and control the required parameters. The data received at the centralized controller or server is analyzed to enhance the efficiency of industrial systems [94]. With highly anticipated developments in the fields of artificial intelligence, data analytics and Blockchain, there is immense potential for Smart industry infrastructure deployments to achieve the emerging paradigms of Factory as a Service (FaaS), Machine as a Service (MaaS), Equipment as a Service (EaaS) and others. Most of the sensing applications require wireless access to the Internet and connectivity to the cloud. IoT is dependent on diverse communication technologies, viz. Wi-Fi, ZigBee, Bluetooth, RFID, Cellular, LPWANs, 5G and others. It is employed in distinct networks and layered structures where connectivity is the key issue. When these technologies are used in an integrated manner in industrial scenarios, connectivity between sensing devices and Internet servers, service reliability and productivity improves. These multi-technology hybrid networks are particularly relevant for complex applications which require different IoT protocols. Each technology collects the data from devices and nodes located in their coverage areas and processes at the corresponding base or access stations. The coordinator or gateway nodes can communicate the received data to the core network and cloud. In such mixed architectures, the associated network server or core network entities perform device management functions such as registration, authentication, resource allocation and data traffic management to the devices connected to their network. Lin et al. [95] used AoA-based Wi-Fi impressions by designating a stepped measurement plot to estimate the user's location in an IIoT environment. It delivers competent accuracy and conversion ability to real-time application IIoT scenarios. Location recognition scheme incorporated with manufacturing meets the necessities of larger resilience and shorter production cycles.

Consequently, it helps to correlate the situation-based applications to the position-based services. Zero message quality-based communication into the industrial systems is proposed in [96] for different sensing applications. The approach improves reliability, but it is not suitable for a wide range of IIoT frameworks. The sensor nodes lying close to the gateway generally consume more energy. These gateway drains early or may face temporal death as they are involved in forwarding packets received from many end nodes, affecting the network's lifetime. A topology control algorithm proposed in [97] is based on binary grey wolf optimization to reduce topology by preserving network connectivity. It uses the active and inactive sensor nodes' schedule in binary format. It introduces a fitness function to minimize the number of active nodes for achieving the target of lifetime expansion of the nodes and network.

## 6. Evaluation Metrics for Localization Techniques

A precise location estimate is an essential service in IoT-based real-time applications. To validate localization algorithms, their performances have to be evaluated using standard measures that fulfill the requirements and limitations of an area where sensor nodes have to be deployed. In this section, evaluation metrics suitable for analyzing and evaluating all types of localization algorithms are discussed.

### 6.1. Accuracy

The accuracy of the localization algorithm indicates that how closely the estimated location coincides with the actual ground truth location of the nodes. Efficient algorithms

offer maximal coincidence. Nevertheless, location precision is not only the prevailing objective of every localization scheme, but it is also reliant on application. The structure of node deployment affects the coherence of desired location precision. Location precision is determined by using average location error as defined,

$$RE = \frac{\sum_{l=1}^{N} \sqrt{(u_l - \overline{u}_l)^2 + (v_l - \overline{v}_l)^2 + (w_l - \overline{w}_l)^2}}{N} \tag{3}$$

where $(u_l, v_l, w_l,)$ and $(\overline{u}_l, \overline{v}_l, \overline{w}_l)$ are true and estimated positions, respectively.

Root mean square localization error is determined by using

$$RMSLE = \sqrt{\sum_{l=1}^{N} \frac{(u_l - \overline{u}_l)^2 + (v_l - \overline{v}_l)^2 + (w_l - \overline{w}_l)^2}{N_L}} \tag{4}$$

where $N_L$ is a number of nodes localized. Along with location error, corresponding geometry projected by the localization algorithms is of equal importance. Parashar et al. [98] have proven that the few localization algorithms with acceptable average localization error show massive variation in relative geometry precision for the assessed and actual network. Consequently, defined new metric called Global Distance Error (GDE) as given by

$$GDE = \frac{1}{D} \sqrt{\frac{\sum_{l=1}^{N} \sum_{m-l+1}^{N} \left( \frac{x_{lm} - \overline{x}_{lm}}{x_{lm}} \right)^2}{\frac{N(N-1)}{2}}} \tag{5}$$

where $x_{lm}$, $\overline{x}_{lm}$ represents the distance between node $l$ and $m$ for actual and positions, respectively. $D$ is the mean transmission range of a sensor node.

### 6.2. Cost

Cost is defined as the expensiveness of an algorithm using power consumption, communication overhead, anchor to node proportion, time complexity, etc. If the primary objective is to maximize network lifetime, then the localization algorithm which minimizes multiple cost parameters is preferable. Nevertheless, cost and accuracy must be balanced as per the needs of the application environment. Few standard cost parameters are described below.

### 6.3. Anchor to Node Ratio

Curtailing the number of anchor nodes is necessary to reduce the deployment cost and increase network lifetime. A large number of anchor nodes in the network use GPS for their location estimate, which is uneconomical and consumes more power. Ultimately, it reduces the lifetime of the network. Therefore, an appropriate proportion of anchor nodes and sensor nodes is crucial in the designing of a localization process. This measure is suitable to compute the trade-off between location precision and the proportion of the localized sensor nodes compared to the deployment cost. The localization algorithm should essentially target a minimal number of anchor nodes to attain the expected precision of an application.

### 6.4. Communication Expenses

Radio communication utilizes maximum power in comparison with the total power consumption of a wireless sensor node. Therefore, reducing communication expenses is essential to maximizing the network lifetime. Scaling is used to optimize communication expenses.

### 6.5. Realization Cost

Generally, every system requires a realization cost for its execution. For the localization schemes, overall incidentals are dispersed for communicating and the computational

time. The communication cost continually impacts information exchange between the sink node, source node and the central regulating component throughout the localization process. The computational time represents the processing cost that arises in the network databases and at the terminal. Computational time is well associated with the expected dimension accuracy of the scheme. Consequently, choosing a suitable localization scheme must be essentially conciliated among the expected dimension accuracy and a reasonable computational cost to reduce the realization costs.

*6.6. Convergence Time*

Convergence time is the time taken for localizing every node in the network. Network size influences the convergence time, and hence, the convergence rate of the localization algorithm has to be analyzed with an increase in the network size. However, for specific applications involving a fixed number of nodes, convergence time is crucial as well. If the location precision of any algorithms is exceptionally high at the cost of longer localization time, it proves an algorithm's impracticality for that scenario. Additionally, for networks with moving nodes, the slower algorithm may fail to imitate the present structure of the network.

*6.7. Energy Consumption*

Power consumption is a crucial problem for IoT-enabled WSN, and researchers propose various approaches to address and manage it. Optimized energy consumption improves the network lifetime and efficiency as well. Average energy consumed is the sum of total energy consumed for sensing and transmission by each node for every round, calculated by using,

$$Average\ Energy\ Consumption = \frac{1}{N}(I_l - R_l) \tag{6}$$

where $I_l$ and $R_l$ are initial and residual energy of node $l$, respectively.

*6.8. Coverage*

Coverage represents the proportion of the nodes which can be localized over the nodes positioned in the network, irrespective of the location accuracy. Node density, the ability of nodes to connect, and anchor nodes' position are the parameters influencing coverage in the network. To evaluate the localization algorithm for coverage, it must be tested for the distinct scenarios, viz. a diverse number of anchor nodes, different network dimensions, and distinct communication range. For the lower node densities, coverage may be less for localization algorithms with arbitrary topology due to connectivity issues.

The performance comparison of evaluation metrics depicting their characteristics are as given in Table 4.

**Table 4.** Performance Comparison of Evaluation Metrics.

| Measurement Technique | Type of Algorithm | Accuracy | Cost | Anchor to Node Ratio | Communication Expenses | Realization Cost | Convergence Time | Energy Consumption | Coverage |
|---|---|---|---|---|---|---|---|---|---|
| Range-based | AoA | Medium | High | High | High | High | High | High | Low |
| | ToA | High | High | High | High | High | Medium | High | Low |
| | TDoA | High | High | Medium | High | High | Medium | High | Medium |
| | RSSI | High | Low | Low | Low | Medium | Medium | High | High |
| Range-Free | Hop Count | Low | Low | Low | Low | Low | Low | High | High |
| | Centroid | Low | Low | Low | Low | Low | Medium | Low | High |
| | APIT | Medium | Low | Medium | High | High | Medium | High | Medium |
| | Analytical Geometry | Low | Low | Medium | Medium | Medium | Medium | High | Medium |
| | Mobile Anchor Node | High | High | High | High | High | Low | Low | Low |

**Table 4.** *Cont.*

| Measurement Technique | Type of Algorithm | Accuracy | Cost | Anchor to Node Ratio | Communication Expenses | Realization Cost | Convergence Time | Energy Consumption | Coverage |
|---|---|---|---|---|---|---|---|---|---|
| Device-based | Wi-Fi | Medium | Low | Low | Low | Medium | Medium | High | Medium |
| | Bluetooth | High | Medium | Medium | High | High | Medium | High | Medium |
| | Camera | High | High | High | High | High | Medium | High | Low |
| | Acoustic | Medium | Medium | Medium | Low | Medium | Medium | Medium | Medium |
| | WEB | High | Low | Low | Low | Medium | Medium | Medium | Medium |
| | RFID | Medium | Low | Medium | Low | Medium | Medium | Medium | Low |
| Device-Free | Infrared | Low | High | High | High | High | Low | Low | Low |
| | Magnetic Sensors | Low | Medium | Low | Low | Low | Medium | Low | Low |
| | WEB | High | Low | Low | Low | Low | Low | Low | High |
| | RFID | High | Low | Low | Low | Low | Low | Low | High |
| | Wi-Fi | Medium | Low | Low | Low | Low | Low | Low | Medium |

## 7. Conclusions

Localization of Internet of Things (IoT) nodes is one of the crucial challenges for several applications where continuous or periodical information regarding accurate location is required. Such applications require the precise location of nodes in real-time and with low energy consumption and minimal cost. The performance requirements in localization also vary from application to application. Explosive growth in nodes with mobility features poses significant localization challenges because of heterogeneity in nodes, technologies, networks and performance requirements. This paper has provided an extensive study of various localization techniques and classified them based on centralized, distributed, iterative, range-based, range-free, device-based, device-free and their subtypes. The paper also discussed the problems, challenges and various technologies and approaches available. Localization applications for a smart city such as services, infrastructure, mobility, transport and health are also discussed. The advantages and limitations of these techniques, along with their comparison, are discussed. All the important metrics used to evaluate the performance of localization techniques are discussed and compared. Future research should focus on precise, faster and energy-efficient localization for broader coverage with lower cost. There are great opportunities to develop artificial intelligence-based collaborative hybrid localization techniques to achieve the application requirements.

**Author Contributions:** All the authors have contributed equally towards the conceptualization, methodology, analysis, investigation, and resources. The original draft and changes were also performed by all the authors. All authors have read and agreed to the published version of the manuscript.

**Funding:** This research received no external funding.

**Data Availability Statement:** Not Applicable, the study does not report any data.

**Conflicts of Interest:** The authors declare no conflict of interest.

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
