# Peer review of "Survey of Localization for Internet of Things Nodes: Approaches, Challenges and Open Issues"

_futureinternet, doi:10.3390/fi13080210_

Round 1
Reviewer 1 Report
This paper classifies and discusses various state-of-the-art techniques proposed for IoT node localization. the different performance metrics that can be used for the localization, comparison of the different techniques, some of the prominent applications in smart cities, and future directions is also covered.
The paper presents an in-depth review of range-free and range-based localization techniques and related concepts.
The authors must be included references for the next paragraphs
Recently, the researchers have started the study on localization in IoT networks for numerous applications. (page 1 – line 40-42).
It is important to introduce the most significant algorithms or references to works in the paragraph .¨
. The central coordinating node analyses the information and estimates positions of different sensor nodes using some algorithm and then conveys it to the individual nodes”. Line 79-80
The localization algorithms for IoT networks are dependent on a variety of measurement techniques.Line 162
The angle of arrival (AoA) measurement technique is called the path of arrival or orientation measurement. The AoA is determined either by using the amplitude response of receiver antennas or the phase response of receiver antennas. Line 177-179
In graph 2, a description could be included to clarify under which characteristics, such as distance, the nodes interact. Line 114
Node density in the network does not affect the performance. The computational complexity of centroid-based techniques is very low… is necessary define Boundary level (log, n, exponential?
The table 2 (line 360) Related to Comparison of Measurement Techniques and the table 3 Summary of device-based and device-free localization techniques. (line 552) are a god job from the authors.
the authors should include a summary table for section 6 Evaluation Metrics for Localization Techniques
expand the conclusions section, for this section 7, is very small
Mandatory review of the work with turnitin software after adjustments, I refer to the reference checking process in this evaluation.
Reviewer 2 Report
The manuscript presents the “Survey of Localization for Internet of Things Nodes: Approaches, Challenges and Open Issues”. There are several suggestions that would immediately improve the readability of the paper and are severely lacking in the current manuscript.
1. Some grammatical mistakes are found such as in abstract line 23 “future directions is also covered.” Should be “future directions are also covered.”. The authors should remove all grammatical mistakes when revise this manuscript.
2. Make the abstract better and highlight more on the topic.
3. Add one paragraph at the end of the introduction part to show the organization of the paper.
4. The author should add a "Related Work" section after the introduction section. And compare the previous studies in the table.
5. The authors should discuss the findings in one paragraph.
6. Authors should revise better and more the current literature in the field.
7. Add challenges and open issues in separate paragraphs.
Round 2
Reviewer 2 Report
The authors have shown efforts in revising their manuscript as well as sufficiently responding to my comments. So, in my opinion, I think this revised version can be considered for publication.